# Recently Discovered Secondary Metabolites from *Streptomyces* Species

**DOI:** 10.3390/molecules27030887

**Published:** 2022-01-28

**Authors:** Heather J. Lacey, Peter J. Rutledge

**Affiliations:** 1School of Chemistry, The University of Sydney, Camperdown, Sydney, NSW 2006, Australia; 2Microbial Screening Technologies, Smithfield, Sydney, NSW 2164, Australia

**Keywords:** biosynthesis, macrolides, natural products, peptides, polyketides, secondary metabolites, *Streptomyces*, structure elucidation, terpenoids

## Abstract

The *Streptomyces* genus has been a rich source of bioactive natural products, medicinal chemicals, and novel drug leads for three-quarters of a century. Yet studies suggest that the genus is capable of making some 150,000 more bioactive compounds than all *Streptomyces* secondary metabolites reported to date. Researchers around the world continue to explore this enormous potential using a range of strategies including modification of culture conditions, bioinformatics and genome mining, heterologous expression, and other approaches to cryptic biosynthetic gene cluster activation. Our survey of the recent literature, with a particular focus on the year 2020, brings together more than 70 novel secondary metabolites from *Streptomyces* species, which are discussed in this review. This diverse array includes cyclic and linear peptides, peptide derivatives, polyketides, terpenoids, polyaromatics, macrocycles, and furans, the isolation, chemical structures, and bioactivity of which are appraised. The discovery of these many different compounds demonstrates the continued potential of *Streptomyces* as a source of new and interesting natural products and contributes further important pieces to the mostly unfinished puzzle of Earth’s myriad microbes and their multifaceted chemical output.

## 1. Introduction

Analysis of *Streptomyces* derived natural product discoveries from 1947 to 1997 indicates that, in terms of the number of publications, the discipline peaked in the late 1960s before tapering off through the 1970s and 1980s and significantly decreasing in the 1990s [1]. Factors contributing to this decline include the development of combinatorial chemistry and associated high-throughput screening (HTS) techniques, a reduction in funding for traditional drug discovery pipelines, and the greater profitability of quality-of-life drugs. Had new bioinformatics technologies and methods such as genome mining not emerged to reinvigorate the field, the declining trend could have seen a cessation of work to exploit *Streptomyces* as a source of new bioactives by 2020 [1]. This is in spite of studies that conservatively estimate that the genus can synthesise some 150,000 more antimicrobial compounds than those currently known, suggesting that *Streptomyces* is far from an exhausted resource [1,2].

A Clarivate Web of Science search was undertaken to gauge the recent level of documented drug discovery research activity in the genus [3], using the terms *Streptomyces*, *isolation*, *new compound* and *novel compound* in the All Fields parameter of the Web of Science search engine. A review of the number of publications dedicated to novel natural product discovery from the *Streptomyces* genus over the last five years reveals a reasonably consistent level of publication on the subject (Figure 1). Comparing the raw numbers of this survey with the Watve study (2001) shows that the current levels of research output (in terms of publication numbers) are comparable to those at the start and end of the Golden Age of natural products [1]. This suggests that *Streptomyces* natural products research publications did not dissipate by 2020 as earlier trajectories may have indicated, nor has there been a major upswing in publications per year in the last five years [1,3]. A proportion of the publications returned in the Web of Science search (on average 7% per year) were excluded from this count because the research they describe did not yield verified novel compounds as isolated secondary metabolites of a *Streptomyces* species. Similarly, reports of partial characterisation were also excluded as the natural products described therein could not be verified as novel. This indicates that a literature count with strict eligibility criteria likely underestimates the true interest in *Streptomyces* drug discovery by focusing on verified novel compound elucidation, rather than simply research activity.

An insight analysis was conducted on scientific reports pertaining to the discovery of new *Streptomyces* secondary metabolites published in 2020. The search returned 43 references relevant to the search terms. Filtering resulted in 32 research papers that met the requirements of the review.

Each of these reports described a unique *Streptomyces* species or strain, and a total of 74 novel compounds were reported across the 32 papers surveyed. Further analysis revealed that marine and soil environments were the equal most common isolation locations, with 56% of these publications citing a *Streptomyces* strain isolated from either a marine environment or terrestrial soil (Figure 2). Notably, a significant proportion (19%, n = 6) of publications did not specify the environmental source of the *Streptomyces* strain; where this occurred, the publication often focused on high performance liquid chromatography-UV-diode array detection (HPLC-DAD) [4], LC-MS [5], or genome mining [6,7] as a drug discovery guidance strategy.

The preliminary analysis of novel secondary metabolites produced by these *Streptomyces* strains also revealed that the largest proportion (39%) of compounds was isolated from marine-derived *Streptomyces*, followed by terrestrial soil (27%) (Figure 3). A further eleven secondary metabolites were isolated from unspecific environments. The compounds reported possess a wide range of chemical scaffold variability and include cyclic and linear peptides, terpenoids, macrolactams, macrolides, glycosides, polyaromatics, and linear polyketides.

Though *Streptomyces* species no longer generate the same level of research output as at their peak in the 1960s, the genus has continued to be the subject of interest and research activity. New genome mining strategies in addition to classic bioprospecting and bioassay-guided isolation have contributed to the plethora of unique chemical scaffolds that continue to characterise both the *Streptomyces* genus and natural products more generally, as an important source of drug leads.

## 2. Cyclic Peptides

In 2020, ten novel cyclic peptides were reported across seven publications [4,6,8,9,10,11,12]. The size of the peptide core ranged from two to seven amino acids, though three of these compounds, ulleungamide C (**1**) [11] and viennamycins A and B (**2** and **3**) (Figure 4) [8], are depsipeptides, and two more, NOS-V1 (**4**) and NOS-V2 (**5**) (Figure 5), possess the typical di-macrocycle composition of a thiopeptide nosiheptide [6].

These compounds originate from both non-ribosomal peptide synthetases (NRPSs) and ribosomal synthesis with post-translational modification. The most common bioactivity observed for this group of compounds was antibacterial activity, followed by cytotoxicity and antifungal proliferation. Pentaminomycin C (**7**) showed activity against Gram-positive bacteria down to 16 μg mL^−1^ [4], but no activity against Gram-negative bacteria. In comparison, NOS-V1 (**4**) and NOS-V2 (**5**) showed activity against Gram-positive and Gram-negative cultures with a more potent effect against Gram-positive than Gram-negative bacteria [6].

The cyclic dipeptide l-leucyl-l-proline (**6**) showed activity against a range of bacterial and fungal targets in inhibition zone studies [10]. The antibacterial activity of the viennamycins **2** and **3** was identified in tests against *S. aureus* 209P and *B. cereus* IP5812, in which these compounds were found to have comparable activity to daptomycin and vancomycin, though no activity was identified against Gram-negative bacteria [8]. Ulleungamide C **1** displayed cytotoxic activity in vitro. In contrast, in vitro studies observed the protective effects of pentaminomycins C (**7**) and D (**8**) (Figure 5) against menadione-induced cytotoxicity [9].

## 3. Peptide Derivatives and Linear Peptides

Three linear peptidic compounds were reported in three publications during 2020. Spongiicolazolicins A (**10**) and B (**11**) (Figure 6) are ribosomally synthesised and post-translationally modified azole-containing peptides isolated from the marine *Streptomyces* sp. CWH03 (NBRC 114659) [13]. Comprising twenty-nine (spongiicolazolicin A) and twenty-eight (spongiicolazolicin B) amino acids, the structures were deduced from 2D NMR, MS/MS experiments, and the sequence of the biosynthetic gene cluster (BGC). Despite studies of similar compounds, such as microcin B17 [14], finding them to have antibacterial bioactivity, biological assay data did not indicate similar activity for the spongiicolazolicins against Gram-positive or Gram-negative species.

Legonimide (**12**, Figure 7) is an indole alkaloid composed of modified phenylalanine and tryptophan amino acids, and was isolated from soil-derived *Streptomyces* sp. CT37 [15]. The authors proposed that this compound is biosynthesised via a condensation reaction between phenyl-acetyl-CoA and indole-3-acetamide. The structure of legonimide was determined from a combination of 1D and 2D NMR data, and biological profiling showed that the compound has moderate antifungal activity against *Candida albicans* (MIC 21.5 μg mL^−1^). The final peptide derivative reported in the review period, 2,3-dihydroxy-1-(indolin-3-yl)butan-1-one (**13**, Figure 7), was isolated from *S. triticiradicis* sp. strain NEAU-H2^T^ [16], collected from the rhizosphere of wheat (*Triticum aestivum* L.). This strain displayed antifungal effects against the mycelial growth of phytopathogenic fungi, though this activity could not be linked to the novel compound in subsequent tests [16].

## 4. Linear Polyketides

An HTS experiment looking for inhibitors of coenzyme A biosynthesis led to the isolation and spectroscopic characterisation of six new phenolic lipids, adipostatins E–J (**14**–**19**), from the marine-derived *S. blancoensis* (Figure 8) [17]. These novel compounds showed activity against both Gram-positive and Gram-negative bacteria: *Enterococcus coli*, *E. faecalis*, *Bacillus subtilis*, *B. anthracis*, *Listeria monocytogenes*, and *Streptococcus pneumoniae*. In contrast, the compounds had low to no effect on *Staphylococcus aureus*, *Salmonella enterica*, *Shigella flexneri*, or *Klebsiella pneumoniae*. The differential in vitro activity of adipostatins E–J suggests that the biological activity is mediated by the chain length and terminal branching.

Similarly, bioactivity screening for tyrosinase inhibitors led to the serendipitous identification of trichostatic acid B (**20**) from *Streptomyces* sp. CA-129531 as a co-metabolite of the target molecules (Figure 9) [18]. Notably, trichostatic acid B did not display any cytotoxic activity up to 50 mg mL^−1^. Nor was any antityrosinase activity detected for trichostatic acid B in a mushroom tyrosinase bioassay, despite trichostatin A (**21**) showing strong inhibitory activity [18], suggesting that the hydroxylamine and/or tertiary methyl group is important to its biological activity. Two additional bioassay-driven natural product discoveries were chresdihydrochalcone (**22**) and chresphenylacetone (**23**, Figure 9), isolated from *S. chrestomyceticus* BCC 24770 [19]. Their molecular structures were spectroscopically characterised, and chresdihydrochalcone was assessed for haemolytic, cytotoxic, and antibacterial activity. Notably, the compound showed moderate activity against methicillin-resistant *S. aureus* (MRSA, MIC 25 μg mL^−1^) and a mutant *E. coli* strain, and stronger activity against *S. aureus* (MIC 18 μg mL^−1^) and *B. subtilis* (MIC 10 μg mL^−1^).

From a group of six compounds isolated from *Streptomyces* sp. MJM3055 and spectroscopically characterised, two compounds, (3*E*,8*E*)-1-hydroxydeca-3,8-dien-5-one (**24**) and (*S,E*)-3-hydroxy-5-oxodec-8-en-1-yl acetate (**25**), were identified as novel natural products via a Jurkat cell antiproliferation screening study (Figure 9) [20]. Using LCMS-guided screening of *Streptomyces* sp. W3002, Lee et al. purified and characterised the new deoxy-streptimidone compounds **26** and **27** (Figure 9) along with two glutarimide ring-opened derivatives of **27** [5]. The new compounds, 3-(5,7-dimethyl-4-oxo-6*E*,8-nonadienyl)-glutarimide (**26**) and 3-(5,7-dimethyl-4-oxo 2*E*,6*E*,8-nonatrienyl)-glutarimide (**27**), possess the characteristic glutarimide ring and alkyl chain and showed no cytotoxic activities against human cervical carcinoma, human hepatoma, or myeloid leukemia up to 100 μg mL^−1^ [21]. Analysis of the putative BGC of both glutarimide derivatives revealed that apart from module five, which contains the C-terminal thioesterase domain, the other four of the first five modules showed a strong similarity to those of the BGC that encodes 9-methylstreptimidone in *S. himastatinicus* [22]. It was proposed that the instability of the dehydratase accessory reactions catalysed by module three was responsible for the absence of the hydroxyl group from the alkyl chain (when these new structures are compared to 9-methylstreptimidone), while an oxidoreductase transforms **26** to **27** [5].

## 5. Terpenoids

Ten novel terpenoid natural products were isolated from four *Streptomyces* species in 2020. Notably, half of these compounds are napyradiomycin derivatives. Napyradiomycins are a large group of meroterpenoids that incorporate a halogenated semi-naphthoquinone moiety. This group of over fifty compounds is divided into three sub-groups based on side-chain characteristics: type A napyradiomycins possess a linear terpenoid chain, type B have a cyclohexane ring, and type C possess a 14-membered ring [23]. In the course of reassessing secondary metabolite biosynthesis by marine-derived *Streptomyces*, Carretero-Molina et al. identified four compounds from *Streptomyces* sp. CA-271078: one each of types A (**28**) and C (**31**), and two of which were type B napyradiomycins (**29** and **30**) (Figure 10) [23]. These compounds were spectroscopically characterised and their stereochemistry was inferred from biosynthetic relationships with previously reported napyradiomycins. These compounds all displayed weak antibacterial activity against MRSA and cytotoxicity to human liver carcinoma cell lines.

The fifth compound was also a type A napyradiomycin, and one of ten isolated from marine *Streptomyces* sp. YP127: 16*Z*-19-hydroxynapyradiomycin A1 (**32**) [24]. This structure was flagged as a potential drug lead through an Nrf-2-activating efficacy bioassay, which indicated that this compound would modulate the activity of Nrf-2, an essential component in the expression of antioxidant defence genes.

Three structurally related naphthoquinones were isolated from the marine-derived *Streptomyces* sp. B9173: flaviogeranins B1 (**33**), B (**34**), and D (**35**), the latter two of which are meroterpenoids (Figure 11). These compounds differ in their prenylation pattern and presence of amino and methyl groups on the hydroxynaphthoquinone core. Flaviogeranins B1 and B showed weak to moderate activity against *S. aureus* and *M. smegmatis* as well as human alveolar basal epithelial adenocarcinoma and human cervical cancer cells. In contrast, flaviogeranin D displayed stronger activity against *S. aureus* (MIC 9.2 μg mL^−1^) and *M. smegmatis* (MIC 5.2 μg mL^−1^), the alveolar basal epithelial adenocarcinoma (IC_50_ 0.6 μg mL^−1^) and human cervical cancer cell lines in vitro (IC_50_ 0.4 μg mL^−1^).

The final terpenoids isolated from the *Streptomyces* genera are two cardinane sesquiterpenes (**36** and **37**) isolated from marine *Streptomyces* sp. ST027706 (Figure 12), a plant endophyte taken from the mangrove *Bruguiera gymnorrihiza* [26]. Cardinanes have primarily been isolated from plant and fungal sources, with only two previous examples of the class being isolated from *Streptomyces* [27,28]. Thus, this finding raises the possibility that bacterial fermentation could be used to obtain cardinane-derived pharmaceuticals.

## 6. Polyaromatics

In total, eleven polyaromatic compounds were isolated from six *Streptomyces* species in 2020. Through herbicidal bioassay screenings, Kim et al. identified strong activity associated with *Streptomyces* strain KRA17-580 against the weed *Digitaria ciliaris* [29]. Two compounds were spectroscopically characterised, a cinnoline-4-carboxamide, 580-H1 (**38**), and a cinnoline-4-carboxylic acid, 580-H2 (**39**, Figure 13). In further phytotoxicity bioassays, the carboxamide proved to be less potent than the carboxylic acid, and electrolyte leakage assays were used to verify that the observed herbicidal activity was caused by destruction of the cell membrane [29].

Flavonoids possess a generic structure in which two aromatic rings (A and B) are connected by a heterocyclic pyran. Investigations into bacterial endophytes of the sea sponge *Halichondria panicea* led to the discovery of *Streptomyces* sp. G248 and three lavandulylated flavonoid secondary metabolites: (2*S*,2″*S*)-6-lavandulyl-7,4′-dimethoxy-5,2′-dihydroxylflavanone (**40**), (2*S*,2″*S*)-6-lavandulyl-5,7,2′,4′-tetrahydroxylflavanone (**41**), and (2″*S*)-5′-lavandulyl-2′-methoxy-2,4,4′,6′-tetrahydroxylchalcone (**42**, Figure 13) [30]. Notably, (2*S*,2″*S*)-6-lavandulyl-5,7,2′,4′-tetrahydroxylflavanone showed strong antibacterial activity (IC_90_ 1 μg mL^−1^) against *E. faecalis*, *S. aureus*, *B. cereus*, *P. aeruginosa*, and *C. albicans*, suggesting that the diol on ring B and the presence of an intact pyran are essential to antibacterial activity [30].

While investigating expression of the nybomycin BGC from *S. albus* subsp. *chlorinus* in heterologous host *S. albus* Del14, Rodriquez Estevez et al. discovered the production of a new metabolite, benzanthric acid (**43**, Figure 14) [31]. This new structure comprises a benzoic acid fused to an anthranilate moiety, and a ^13^C feeding study indicated that anthranilic acid is incorporated into **43**, but not into the nybomycin scaffold (**44**). Notably, benzanthric acid was not identified in either the origin strain (*chlorinus*) or the unmodified host strain (Del14). Given the absence of a benzoic acid supply from the nybomycin BGC, Rodriquez Estevez et al. proposed that the benzoic acid is supplied by the host strain, possibly through a phenylalanine degradation pathway, and so too the enzyme that catalyses reaction between the anthranilate moiety and benzoic acid. This report indicated that despite the simultaneous production of nybomycin and benzanthric acid in this strain, there are substantial differences in the biosynthetic routes to the two compounds. The new benzanthric acid compound was tested for phytotoxic activity against *Agrostis stolonfera*, though no activity was identified. This study demonstrates an unexpected interplay between a target BGC and host strain, once integrated.

The second fused tricyclic compound isolated in 2020 was 4-((3*S*,4*R*,5*S*)-3,4,5-trihydroxy-6-(hydroxymethyl) tetrahydro-2Hpyran-2-yloxy)phenazine-1-carboxylic acid (**45**), a phenazine that features a benzoic acid moiety and glucose pendant (Figure 14) [32]. The producing strain *Streptomyces* sp. strain UICC B-92 is an endophyte of the large crane fly *Nephrotoma altissima*. The isolated compound was found to be active against Gram-positive bacteria *S. aureus* and *B. cereus* in disk diffusion assays.

In a successful attempt to highlight the value of chemical exploration of less investigated ecological niches, Voitsekhovskaia et al. isolated anthraquinone derivatives baikalomycins A–C (**46**–**48**, Figure 15) from the mollusc (*Benedictia baicalensis*) endophyte *Streptomyces* sp. strain IB201691-2A [33]. Baikalomycin A was identified as an aquayamycin derivative modified with β-d-amicetose and two additional hydroxyl groups on C-6a and C-12a of the aglycone. Baikalomycin B **47** is a disaccharide derivative of the same aglycone that incorporates epimeric α-l-amicetose as the *O*-linked second sugar. In baikalomycin C, the A ring of the parent aquayamycin structure is opened to afford an anthraquinone (from rings B, C, and D) with a 3-hydroxy-3-methyl butanoic acid side chain at C4a of the parent aglycone, and a disaccharide comprising β-d-amicetose and α-l-aculose at C9. The stereochemistry within the A ring of baikalomycins A (**46**) and B (**47**) and (by extrapolation) the sidechain of **48** was assigned by comparison with previously isolated compounds with similar biosynthetic origin, while the stereochemistry in the B ring of **46** and **47** remains uncertain. The putative type II polyketide synthase (PKS) BGC possesses typical genes responsible for angucycline core assembly, genes necessary for biosynthesis of the deoxy sugar, and three genes encoding the glycosyltransferase enzymes. Weak antibacterial and moderate cytotoxic activity were identified for baikalomycin C, while baikalomycins A and B only showed weak or no cytotoxic activity against various cell lines.

Two further tetracyclic metabolites were isolated from an endophytic *Streptomyces* species: gardenomycins A and B (**49** and **50**) [34]. *S. bulli* GJA1 was extracted from the bark of *Gardenia jasminoides*, and the molecular structures of the secondary metabolites deduced from spectroscopic analysis. Notably, gardenomycins A and B possess an unprecedented ether bridge between their B and C rings. Kim et al. proposed that an ether bridge was formed from ring opening of an α-oriented epoxide on ring C by a hydroxyl group on ring B [34]. These compounds did not display antiproliferative or antivirulence effects in bioassays.

## 7. Macrocycles

Macrocycle discoveries fall into two subgroups: macrolides (macrocyclic esters) and macrolactams (macrocyclic amides). Five macrolides (**51**–**55**, Figure 16) were identified while reactivating the quiescent pladienolide B biosynthetic pathway in a laboratory strain of *S. platensis* AS6200 [35]. Unsurprisingly, the laboratory strain showed high similarity (>95% nucleotide similarity with the published *pld* BGC [36]). One notable difference was that the PKS was encoded by five genes in AS6200, while the previously reported PKS was encoded by four genes [35,36]. The molecular structure of these compounds was deduced from spectroscopic data, and the compounds were tested in antibacterial, antifungal, antiprotozoal, and antiproliferation bioassays. The results indicated that all the congeners have less cytotoxic activity than the parent compound pladienolide B.

Six new analogues of oxazole-bearing macrodiolide conglobatin (**56**) were isolated from the previously unreported Australian *Streptomyces* strain MST-91080 (Figure 17) [37]. Conglobatins B–E (**57**–**62**) vary in the distribution of methyl groups around the macrocyclic core, proposed to arise via variable addition of methylmalonyl-CoA extender units during their biosynthesis. Three of the new conglobatins (B1 **56**, C1 **57**, and C2 **58**) displayed enhanced cytotoxicity against NS-1 myeloma cells, IC_50_ = 0.084, 1.05, and 0.45 µg mL^−1^ respectively, an increase in potency relative to the parent compound (IC_50_ = 1.39 µg mL^−1^).

Five macrolactams were described in two separate reports (Figure 18). Hashimoto et al. identified a gene cluster containing a type-I PKS with glutamate mutase genes in the genome of *S. rochei* IFO12908 [7]. The BGC was expressed in a host strain derived from *S. avermitilis* and produced a polyene macrolactam **63,** which was subsequently named JBIR-156; its stereochemistry was predicted from the biosynthetic pathway. Given the previous reports of cytotoxicity for related macrolactam structures, the compound was tested on a range of mammalian cells and showed strong activity against human ovarian adenocarcinoma (IC_50_ 9.5 μM) and T lymphoma Jurkat (IC_50_ 3.5 μM) cell lines. The final four compounds are polycyclic tetramate macrolactams and were uncovered using a mixture of gene- and growth-condition-manipulation techniques on a cryptic BGC in the genome of *S. somaliensis* SCSIO ZH66 [39]. Somamycin A (**64**) was identified spectroscopically as a polycyclic tetramate macrolactam, with the remaining compounds identified as the 10-*epi*-hydroxymaltophilin (B, **65**), 10-*epi*-maltophilin (C, **66**), and dihydro 10-*epi*-maltophilin (D, **67**) analogues. These compounds were tested against two plant fungal pathogens and mammalian cell lines, showing antifungal activity and cytotoxicity, with somamycin displaying the strongest antifungal activity against *Fusarium oxysporum* and *Alternaria brassicae* (1.56 and 3.12 μg mL^−1^, respectively) and human cancer cell lines HCT116 and K562 (IC_50_ 0.6 and 1.5 μM, respectively) [39].

## 8. Furans

Five *Streptomyces*-derived secondary metabolites containing modified benzofurans were isolated in 2020 from *S. pratensis* strain KCB-132: (±)-pratenone A (**68** and **69**), and furamycins I–III (**70**–**72**, Figure 19) [40]. The core of these compounds is a benzofuranone ring system linked to naphthalene. Pratenone A is distinguished by the presence of a 3-methylnaphthalen-1,7-diol moiety, furamycins I and II possess 3,7-dihydroxy-3-methyl-3,4-dihydronaphthalen-1(2H)-one ring systems, and furamycin III is characterised by a methoxy group in place of the carbonyl on the benzofuran(one) portion. These compounds were tested for antibacterial and cytotoxic activity. Furamycin III showed moderate activity against pancreatic cancer cells (IC_50_ 26.0 μg mL^−1^) and hepatocellular carcinoma (IC_50_ 42.7 μg mL^−1^) in vitro. Only pratenone A displayed antibacterial activity against *S. aureus* (MIC 8 μg mL^−1^) [40].

The final furanic compound reported was 2-alkyl-4-hydroxymethylfuran carboxamide (AHFA, **73**), isolated from a soil-derived *Streptomyces* sp. RK44 [41]. AHFA is a homolog of methylenomycin furan (MMF, **74**) and is distinguished by the presence of an amide moiety at C-3 rather than a carboxylic acid (Figure 19). MMFs share a common furan ring structure and are autoregulators that act in the signalling pathway that stimulates production of methylenomycin biosynthesis [42]. Weak antiproliferative activity was observed (EC_50_ 89.6 μM) when AHFA was tested on a melanoma cell line, though no antibacterial or antiplasmodial activities were detected when tested.

## 9. Conclusions

A review of the natural products isolated from *Streptomyces* in 2020 reveals a diversity of novel chemical structures and a thriving and multifaceted area of drug discovery research. The level of research is on par with the end of the Golden Age of antibiotics on a publications per year basis. During the year, over seventy novel compounds were isolated from *Streptomyces* species using a combination of HPLC-DAD, genomics, and bioassays to guide the isolation of new compounds.

The *Streptomyces* species investigated derive from a wide range of environments, with marine locales being the most investigated, followed by terrestrial soil. Unsurprisingly, this resulted in most new natural products being isolated from marine and soil environments as well. A diverse range of compounds spanning many different chemical classes have been identified, including cyclic and linear peptides, linear polyketides, terpenoids, polyaromatics, macrocycles, and furans. Interestingly, the biological activity identified for these compounds was generally limited, and not tested across a consistent set of pathogenic species or cell lines, leaving an incomplete activity profile for the compound set.

Overall, the discovery of these many and different compounds demonstrates the continued potential of *Streptomyces* as a source of new and interesting natural products and contributes important pieces to the largely unfinished puzzle of Earth’s myriad microbes and their multifaceted chemical output.

## Figures and Tables

**Figure 1 molecules-27-00887-f001:**
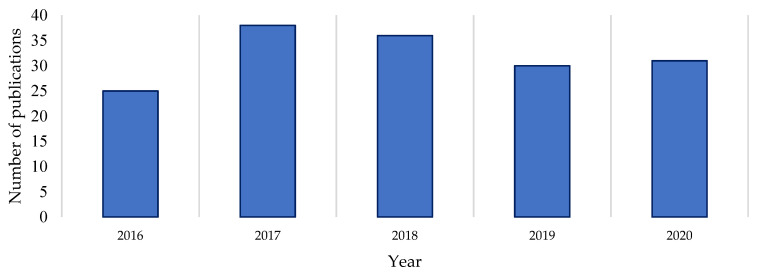
Drug discovery publications 2016–2020 focused on the isolation of *Streptomyces* natural products [3].

**Figure 2 molecules-27-00887-f002:**
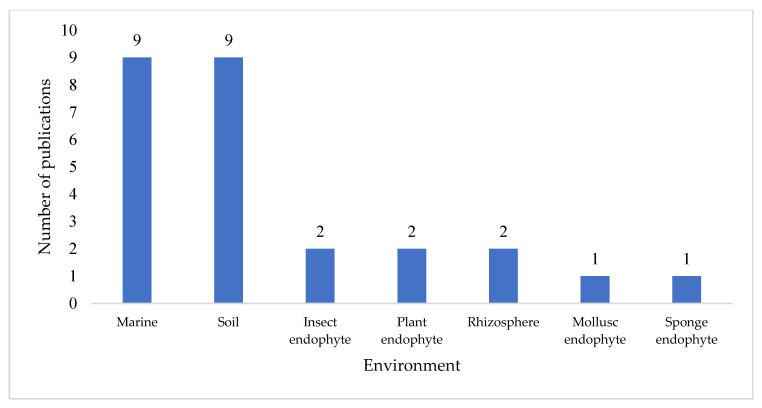
Number of publications for each of the isolation environments of secondary-metabolite-producing *Streptomyces* strains reported in 2020. A further 6 publications did not specify the environmental source of the strain under study.

**Figure 3 molecules-27-00887-f003:**
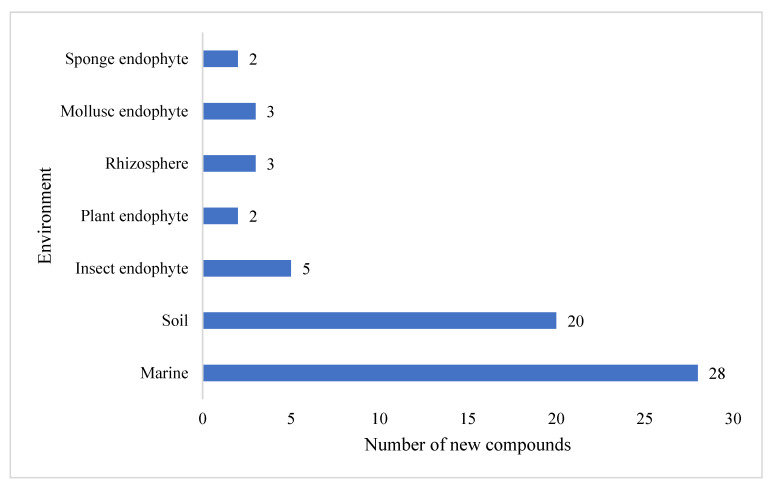
The environmental distribution of *Streptomyces* secondary metabolites isolated in 2020. A further 11 compounds originated from organisms of unspecified origin.

**Figure 4 molecules-27-00887-f004:**
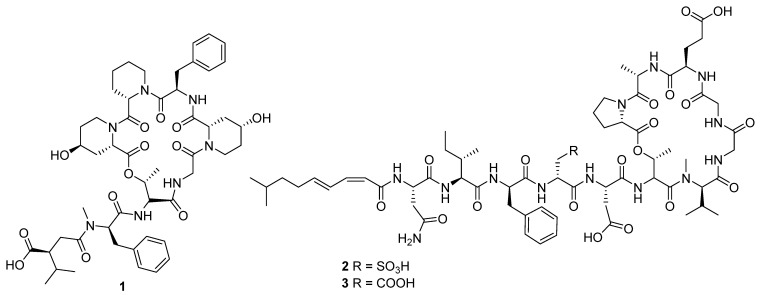
The molecular structures for ulleungamide C (**1**) and viennamycins A and B (**2** and **3**) [8,11].

**Figure 5 molecules-27-00887-f005:**
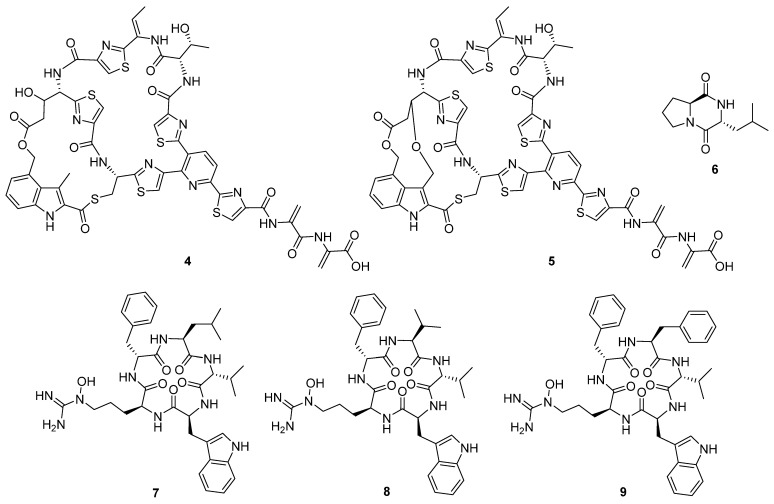
Cyclic peptides NOS-V1 (**4**), NOS-V2 (**5**), cyclo-l-leucyl-l-proline (**6**), and pentaminomycins C–E (**7**–**9**) [6,9,10].

**Figure 6 molecules-27-00887-f006:**
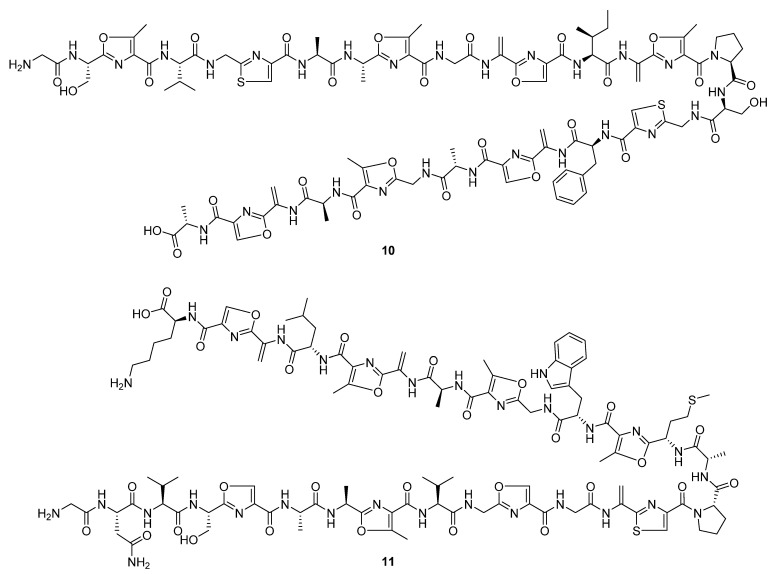
The structures of spongiicolazolicins A (**10**) and B (**11**), ribosomally synthesised and post-translationally modified azole-containing peptides [13].

**Figure 7 molecules-27-00887-f007:**
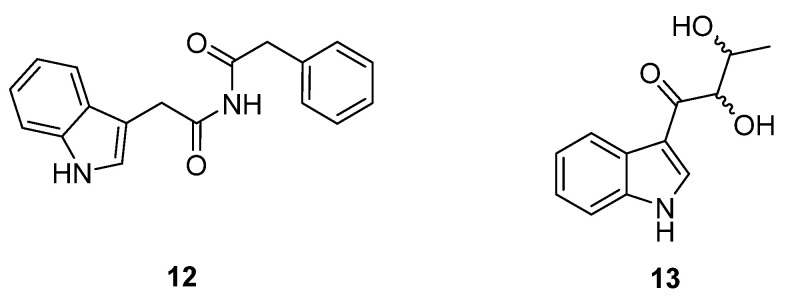
The structures of legonimide (**12**) and 2,3-dihydroxy-1-(indolin-3-yl)butan-1-one (**13**) [15,16].

**Figure 8 molecules-27-00887-f008:**
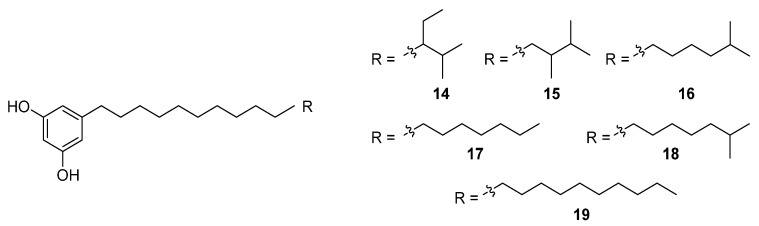
Molecular structures of adipostatins E–J (**14**–**19**) [17].

**Figure 9 molecules-27-00887-f009:**
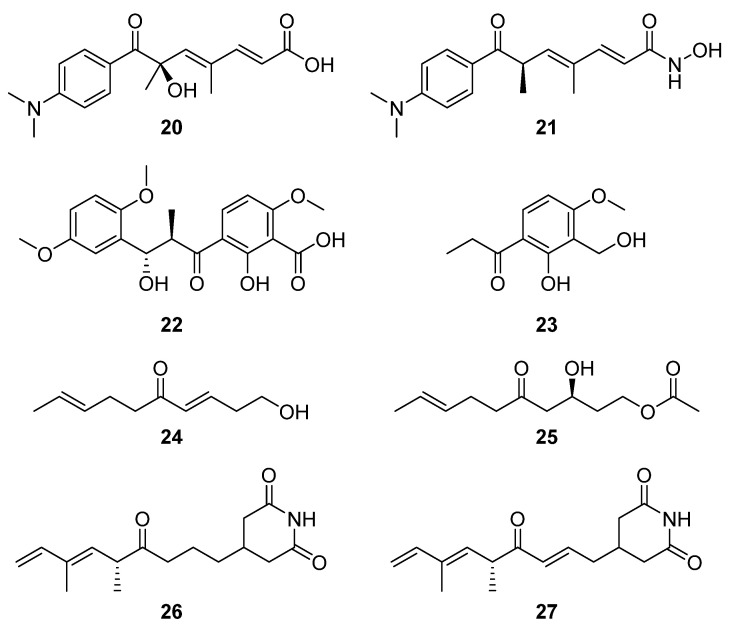
The molecular structures of trichostatic acid B (**20**), trichostatin A (**21**), chresdihydrochalcone (**22**), chresphenylacetone (**23**), (3*E*,8*E*)-1-hydroxydeca-3,8-dien-5-one (**24**), (*S*,*E*)-3-hydroxy-5-oxodec-8-en-1-yl acetate (**25**), 3-(5,7-dimethyl-4-oxo-6*E*,8-nonadienyl)-glutarimide (**26**), and 3-(5,7-dimethyl-4-oxo 2*E*,6*E*,8-nonatrienyl)-glutarimide (**27**) [5,18,19,20].

**Figure 10 molecules-27-00887-f010:**
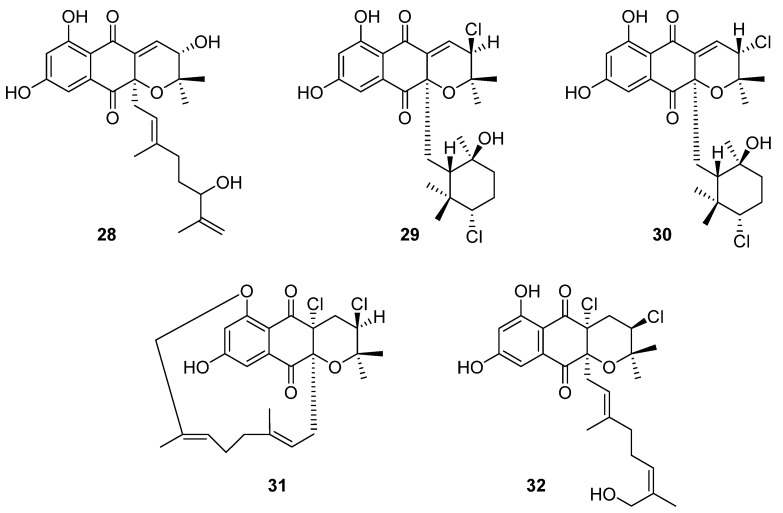
The molecular structures of four unnamed napyradiomycins (**28**–**31**) and 16*Z*-19-hydroxynapyradiomycin A1 (**32**) first reported in 2020 [23,24].

**Figure 11 molecules-27-00887-f011:**
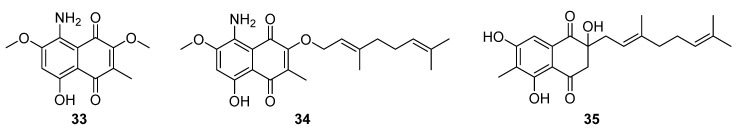
The molecular structures of flaviogeranins B1 (**33**), B (**34**), and D (**35**) [25].

**Figure 12 molecules-27-00887-f012:**
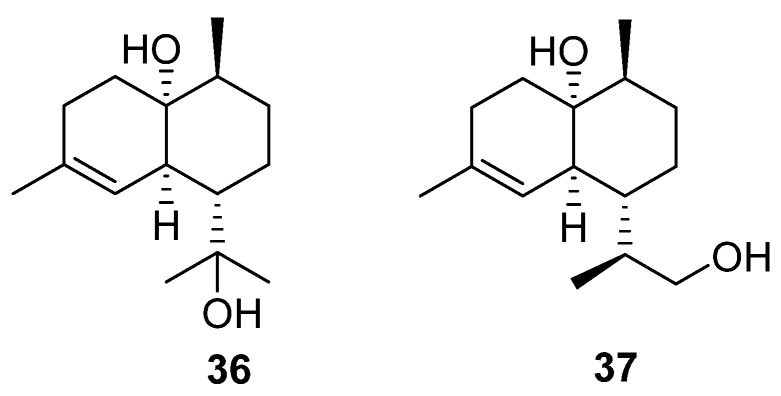
Molecular structures of two cardinanes (**36** and **37**) isolated from an endophytic *Streptomyces* sp. [26].

**Figure 13 molecules-27-00887-f013:**
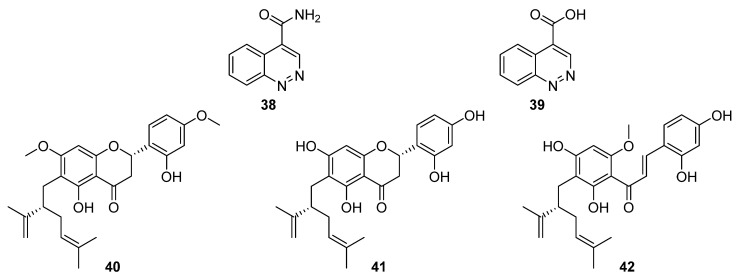
The structure of 580-H1 (**38**), 580-H2 (**39**), (2*S*,2″*S*)-6-lavandulyl-7,4′-dimethoxy-5,2′-dihydroxylflavanone (**40**), (2*S*,2″*S*)-6-lavandulyl-5,7,2′,4′-tetrahydroxylflavanone (**41**), and (2″*S*)-5′-lavandulyl-2′-methoxy-2,4,4′,6′-tetrahydroxylchalcone (**42**) [29,30].

**Figure 14 molecules-27-00887-f014:**
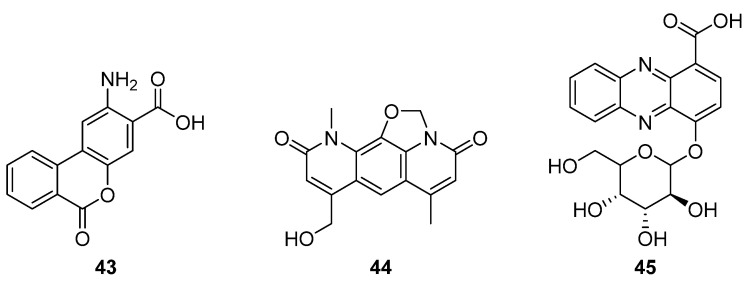
Structures of benzanthric acid (**43**), nybomycin (**44**), and 4-(3*S*,4*R*,5*S*)-3,4,5-trihydroxy-6-(hydroxymethyl) tetrahydro-2Hpyran-2-yloxy)phenazine-1-carboxylic acid (**45**) [31,32].

**Figure 15 molecules-27-00887-f015:**
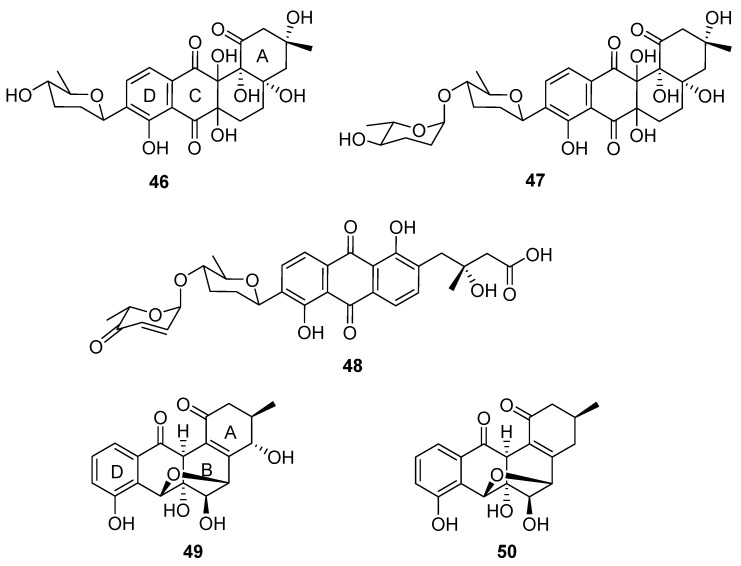
The structures of baikalomycins A–C (**46**–**48**) and gardenomycins A and B (**49** and **50**) [33,34].

**Figure 16 molecules-27-00887-f016:**
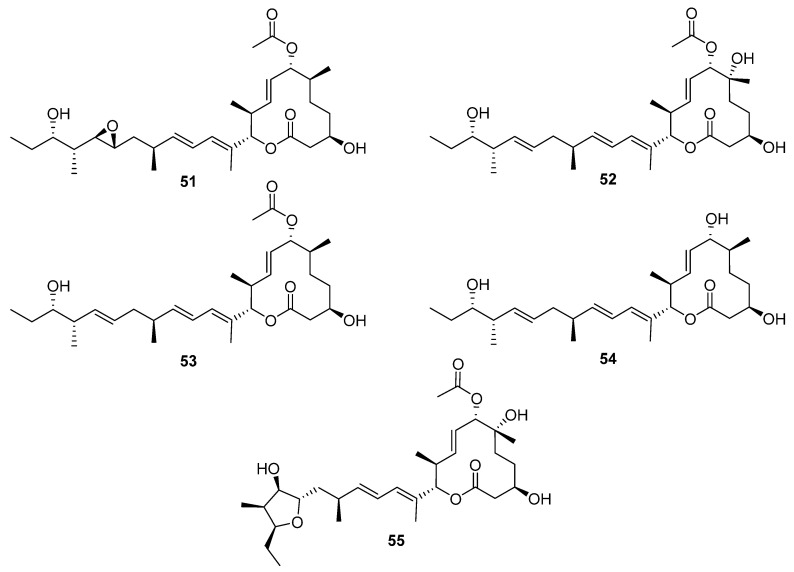
The structures of macrolide natural products **51**–**55** identified while reactivating the pladienolide B biosynthetic pathway in *S. platensis* AS6200 [35].

**Figure 17 molecules-27-00887-f017:**
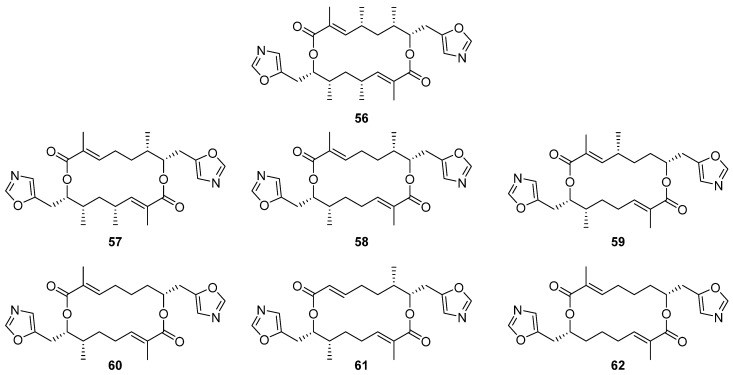
Structures of conglobatins **56**–**62** isolated from *Streptomyces* sp. MST-91080; the parent compound conglobatin (**56**) was previously reported as a secondary metabolite of *Streptomyces conglobatus* ATCC 31005; the others, conglobatins B1 (**57**), C1 (**58**), C2 (**59**), D1 (**60**), D2 (**61**), and E (**62**), were reported for the first time in 2020 [37,38].

**Figure 18 molecules-27-00887-f018:**
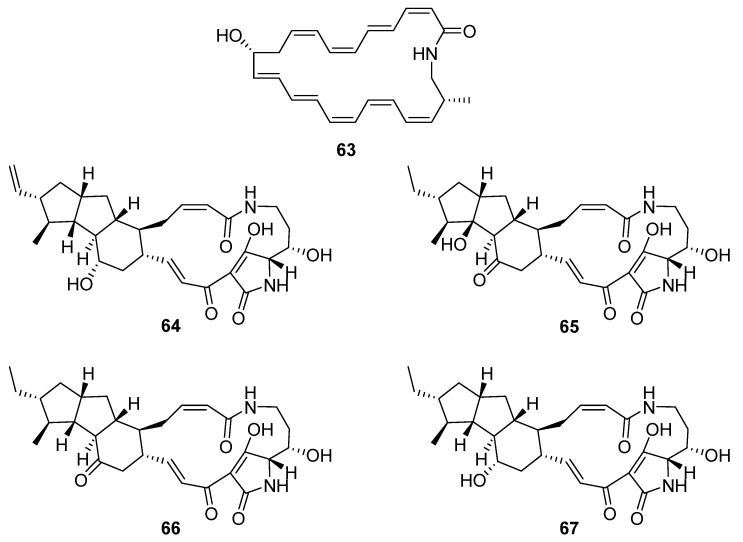
The molecular structures of JBIR-156 (**63**) and somamycins A–D (**64**–**67**) [7,39].

**Figure 19 molecules-27-00887-f019:**
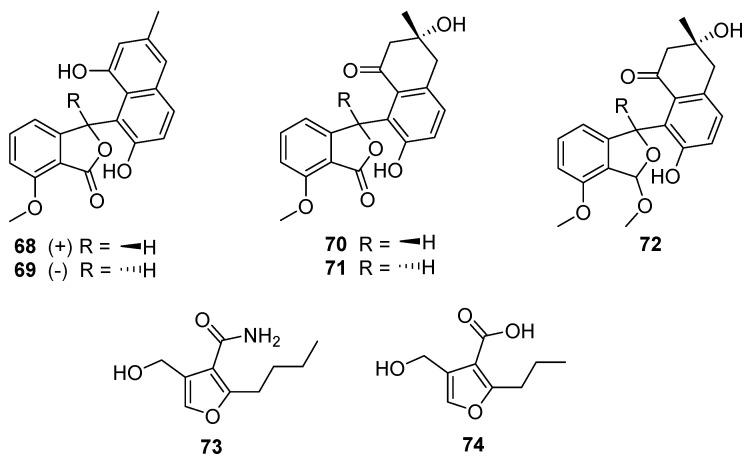
The structures of (±)-pratenone A (**68**, **69**), furamycin I (**70**), furamycin II (**71**), furamycin III (**72**), 2-alkyl-4-hydroxymethylfuran carboxamide (AHFA, **73**), and methylenomycin furan (MMF, **74**) [40,41].

## Data Availability

Not applicable.

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
