# Peer review of "Recently Discovered Secondary Metabolites from Streptomyces Species"

_molecules, 2022, doi:10.3390/molecules27030887_

Round 1

Reviewer 1 Report

This is a useful review and I like the idea of essentially one year's coverage of a topical area. It helps to give a measure of the level of activity as well an update on new compounds of interest to a wide community. There are a number of issues that need to be addressed.

  1. Title rather unspecific. "recent" can mean anything up to 20 years to some people. Why not simply put 2020 in the title.
  2. Combine Figures 5 and 6 - this would aid reading of the relevant discussion.
  3. Line 106. While legonimide may be derived from tryptophan and phenylaline, is there any evidence, particulary for which enantiomer?
  4. Line 118 - define HTS - looks like we may have two "screenings". Also 14-18 should be 14-19.
  5. Line 127 trichostatic acid B (20) - insert number here.
  6. Structure 28 is incorrect.
  7. Compound 33 is NOT a meroterpenoid - no terpene! Structure 34 is incorrect - methylene missing.
  8. Line 173 - should this be tetrahydroxynaphthalene?
  9. Structure 36 is incorrect - OH missing, and remove Ha/Hb.
  10. lines 213-214 - I find it difficult to envisage what biosynthetic steps are shared her?
  11. Compound 25 - is this actually a benzoic acid moiety?
  12. lines 229-231 - tis whole section could be rewritten. Compound 47 is NOT a di-amicetoside. 48 is more than just a ring-opened analogue - aromatisation has occurred, and is the stereochemistry of the side chain actually known?
  13. The acknowledgement is not appropriate.
  14. Overall, a useful article, but the authors should go through the structures carefully to check them. The ones that I have highlighted were a few that looked wrong structurally or biosynthetically  at first glance.

Author Response

We thank the reviewer for their careful reading of our manuscript and helpful suggestions to improve it. Please see the attachment for our point-by-point response. 

Reviewer 2 Report

The manuscript "Recently discovered secondary metabolites from Streptomyces species" is a comprehensive review of recently studies of secondary metabolites of Streptomyces. There are a lot of useful data and descriptions of metabolites, they biological activity in this manuscript. This work may be interesting for researches who study Streptomyces for natural bioactive components production. I think the manuscript may be published in the Molecules journal after minor revision. 1. Searching method for literature analysis should be specified. Which platform was used? Which queries were used? 2. It will be better to summarize information about metabolites, methods of they analysis and biological activity it the table.

Author Response

(The authors gave the same response as above.)
